

# *De novo* genome assembly of *Geosmithia morbida*, the causal agent of thousand cankers disease

Taruna A. Schuelke[1], Anthony Westbrook[2], Kirk Broders[3], Keith Woeste[4] and Matthew D. MacManes[1]

[1] Department of Molecular, Cellular, & Biomedical Sciences, University of New Hampshire, Durham, New Hampshire, United States
[2] Department of Computer Science, University of New Hampshire, Durham, New Hampshire, United States
[3] Department of Bioagricultural Sciences and Pest Management, Colorado State University, Fort Collins, Colorado, United States
[4] Hardwood Tree Improvement and Regeneration Center, USDA Forest Service, West Lafayette, Indiana, United States

## ABSTRACT

*Geosmithia morbida* is a filamentous ascomycete that causes thousand cankers disease in the eastern black walnut tree. This pathogen is commonly found in the western U.S.; however, recently the disease was also detected in several eastern states where the black walnut lumber industry is concentrated. *G. morbida* is one of two known phytopathogens within the genus *Geosmithia*, and it is vectored into the host tree via the walnut twig beetle. We present the first *de novo* draft genome of *G. morbida*. It is 26.5 Mbp in length and contains less than 1% repetitive elements. The genome possesses an estimated 6,273 genes, 277 of which are predicted to encode proteins with unknown functions. Approximately 31.5% of the proteins in *G. morbida* are homologous to proteins involved in pathogenicity, and 5.6% of the proteins contain signal peptides that indicate these proteins are secreted. Several studies have investigated the evolution of pathogenicity in pathogens of agricultural crops; forest fungal pathogens are often neglected because research efforts are focused on food crops. *G. morbida* is one of the few tree phytopathogens to be sequenced, assembled and annotated. The first draft genome of *G. morbida* serves as a valuable tool for comprehending the underlying molecular and evolutionary mechanisms behind pathogenesis within the *Geosmithia* genus.

## INTRODUCTION

Studying molecular evolution of any phenotype is now made possible by the analysis of large amounts of sequence data generated by next-generation sequencing platforms. This is particularly beneficial for the study of emerging fungal pathogens, which are progressively recognized as a threat to global biodiversity and food security. Furthermore, in many cases their expansion is a result of anthropogenic activities and an increase in trade of

Corresponding author
Kirk Broders,
kirk.Broders@colostate.edu

fungal-infected goods (*Fisher et al., 2012*). Fungal pathogens are capable of evolving rapidly in order to overcome host resistance, fungicides, and to adapt to new hosts and environments. Whole genome sequence data are useful in identifying the mechanisms of adaptive evolution within fungi (*Stukenbrock et al., 2011*; *Gardiner et al., 2012*; *Condon et al., 2013*). For instance, *Stukenbrock et al. (2011)* investigated the patterns of evolution in fungal pathogens during the process of domestication in wheat using all aligned genes within the genomes of wheat pathogens. They found that *Zymoseptoria tritici*, a domesticated wheat pathogen (formerly known as *Mycosphaerella graminicola*), underwent adaptive evolution at a higher rate than its wild relatives, *Z. pseudotritici* and *Z. ardabiliae* (*Stukenbrock et al., 2012*). The study also revealed that many of the pathogen's 802 secreted proteins were under positive selection. A study by *Gardiner et al. (2012)*, identified genes encoding aminotransferases, hydrolases, and kinases that were shared between *Fusarium pseudograminearum* and other cereal pathogens. Using phylogenomic analyses, the researchers demonstrated that these genes had bacterial origins. These studies highlight the various evolutionary means that fungal species employ in order to adapt to specific hosts, as well as the importance of genomics and bioinformatics in elucidating evolutionary mechanisms within the fungal kingdom.

Many tree fungal pathogens associate with bark beetles in the family Scolytinae (*Six & Wingfield, 2011*). With climate change, beetles and their fungal symbionts can invade new territory and become major invasive forest pests on a global scale (*Kurz et al., 2008*; *Sambaraju et al., 2012*). A well-known example of an invasive pest is the mountain pine beetle and its symbiont, *Grosmannia clavigera* that has affected approximately 3.4 million of acres of lodgepole, ponderosa, and five-needle pine trees in Colorado alone since the outbreak began in 1996 (*Massoumi Alamouti et al., 2014*; *Colorado State Forest Service, 2015*). Another beetle pest in the western U.S., *Pityophthorus juglandis* (walnut twig beetle), associates with several fungal species, including the emergent fungal pathogen *Geosmithia morbida* (*Tisserat et al., 2009*; *Kolarik et al., 2011*).

Reports of tree mortality triggered by *G. morbida* infections first surfaced in 2009 (*Kolarik et al., 2011*), while the fungus was described as a new species in 2011 (*Tisserat et al., 2009*). This fungus is vectored into the host via *P. juglandis* and is the causal agent of thousand cankers disease (TCD) in *Julgans nigra* (eastern black walnut) (*Zerillo et al., 2014*). This walnut species is valued for its wood, which is used for furniture, cabinetry, and veneer. Although *J. nigra* trees are planted throughout western U.S. as a decorative species, they are indigenous to eastern North America where the walnut industry is worth hundreds of millions of dollars (*Rugman-Jones et al., 2015*; *Zerillo et al., 2014*). In addition to being a major threat to the eastern populations of *J. nigra*, TCD is of great concern because certain western walnut species including *J. regia* (the Persian walnut), *J. californica,* and *J. hindsii* are also susceptible to the fungus according to greenhouse inoculation studies (*Utley et al., 2013*).

The etiology of TCD is complex because it is a consequence of a fungal-beetle symbiosis. The walnut twig beetle, which is only known to attack members of genera *Juglans* and *Pterocarya*, is the most common vector of *G. morbida* (*Kolarik et al., 2011*). Nevertheless, other beetles are able to disperse the fungus from infested trees

(*Kolařík, Kostovčík & Pažoutová, 2007*; *Kolařík, & Jankowiak, 2013*). As vast numbers of beetles concentrate in the bark of infested trees, fungal cankers form and coalesce around beetle galleries and entrance holes. As the infection progresses, the phloem and cambium discolor and the leaves wilt and yellow. These symptoms are followed by branch dieback and eventual tree death, which can occur within three years of the initial infection (*Kolarik et al., 2011*). Currently, 15 states in the U.S. have reported one or more incidences of TCD, reflecting the expansion of WTB's geographic range from its presumed native range in a few southwestern states (*Rugman-Jones et al., 2015*). Additionally, TCD has also been found in Europe where walnut species are planted for timber (*Montecchio & Faccoli, 2014*).

To date, *G. morbida* is one of only two known pathogens within the genus *Geosmithia*, which consists of mostly saprotrophic beetle-associated species (the other pathogen is *G. pallida*) (*Lynch et al., 2014*). The ecological complexity this vector-host-pathogen system exhibits makes it an intriguing lens for studying the evolution of pathogenicity. A well-assembled reference genome will enable us to identify genes unique to *G. morbida* that may be utilized to develop sequence-based tools for detecting and monitoring epidemics of TCD and for exploring the genomic features of *Geosmithia* species, which may help explain the evolution of pathogenicity. Here, we present a *de novo* genome assembly of *Geosmithia morbida*. The objectives of this study are to: 1) assemble the first, high-quality draft genome of this pathogen; 2) annotate the genome to better understand the genomic composition of *Geosmithia* species; and 3) briefly compare the genome of *G. morbida* to two other fungal pathogens for which genomic data are available: *Fusarium solani*, a root pathogen that infects soybean, and *Grosmannia calvigera*, a pathogenic ascomycete that associates with the mountain pine beetle and kills lodgepole pines in North America.

## METHODS

### DNA extraction and library preparation

DNA was extracted using the CTAB method as outlined by the Joint Genome Institute for Genome Sequencing from lyophilized mycelium of *G. morbida* (isolate 1262, host: *Juglans californica*) from southwestern California (*Kohler & Francis, 2015*). The total DNA concentration was measured using Nanodrop, and samples for sequencing were sent to Purdue University Genomics Core Facility in West Lafayette, Indiana. DNA libraries were prepared using the paired-end Illumina Truseq protocol and mate-pair Nextera DNA Sample Preparation kits with average insert sizes of 487 and 1921 bp, respectively. These libraries were sequenced on the Illumina HiSeq 2500 using a single lane with a maximum read length of 101 bp.

### Preprocessing sequence data

To assess the quality of our data, we ran FastQC (v0.11.2) (https://goo.gl/xHM1zf) (*Andrews, 2015*) and SGA Preqc (v0.10.13) (https://goo.gl/9y5bNy) on our raw sequence reads (*Simpson, 2013*). Both tools aim to supply the user with information such as per base sequence quality score distribution (FastQC) and frequency of variant branches in *de Bruijn* graphs (Preqc) that aid in selecting appropriate assembly tools and parameters. The paired-end raw reads were corrected using a Bloom filter-based error correction

tool called BLESS (v0.16) (https://goo.gl/Kno6Xo) (*Heo et al., 2014*). Next, the error corrected reads were trimmed with Trimmomatic, version 0.32, using a Phred threshold of 2, following recommendations from *MacManes (2014)* (https://goo.gl/FFoFjL) (*Bolger, Lohse & Usadel, 2014*). NextClip, version 1.3.1, was leveraged to trim adapters in the mate-pair read set (https://goo.gl/aZ9ucT) (*Leggett et al., 2014*).

### *De novo* genome assembly and evaluation

The *de novo* genome assembly was constructed with ALLPaths-LG (v49414) (https://goo.gl/03gU9Z) (*Gnerre et al., 2011*). The assembly was evaluated with BUSCO (v1.1b1) (https://goo.gl/bMrXIM), a tool that assesses genome completeness based on the presence of single-copy orthologs (*Simão et al., 2015*). We also generated length-based statistics for our *de novo* genome with QUAST (v2.3) (https://goo.gl/5KSa4M) (*Gurevich et al., 2013*). The raw reads were mapped back to the genome using BWA version 0.7.9a-r786 to further assess the quality of the assembly (https://goo.gl/Scxgn4) (*Li & Durbin, 2009*).

### Structural and functional annotation of *G. morbida* genome

We used the automated genome annotation software Maker version 2.31.8 (*Cantarel et al., 2008*). Maker identifies repetitive elements, aligns ESTs, and uses protein homology evidence to generate ab initio gene predictions (https://goo.gl/JiLA3H). We used two of the three gene prediction tools available within the pipeline, SNAP and Augustus. SNAP was trained using gff files generated by CEGMA v2.5 (a program similar to BUSCO) (*Parra, Bradnam & Korf, 2007*). Augustus was trained with *Fusarium solani* protein models (v2.0.26) downloaded from Ensembl Fungi (*EnsemblFungi, 2015*). In order to functionally annotate the genome, the protein sequences produced by the structural annotation were blasted against the Swiss-Prot database, and target sequences were filtered for the best hits (*Swiss-Prot, 2015*). A small subset of the resulting annotations was visualized and manually curated in WebApollo v2.0.1 (*Lee et al., 2013*). The final annotations were also evaluated with BUSCO (v1.1b1) (https://goo.gl/thTGzH).

### Assessing repetitive elements profile

To assess the repetitive elements profile of *G. morbida*, we masked only the interspersed repeats within the assembled scaffolds with RepeatMasker (v4.0.5) (https://goo.gl/TXrbr3) (*Smit, Hubley & Green, 1996*) using the sensitive mode and default values as arguments. In order to compare the repetitive element profile of *G. morbida* with *F. solani* (v2.0.29) and *G. clavigera* (kw1407.GCA_000143105.2.30), the interspersed repeats of these two fungal pathogens were also masked with RepeatMasker. The genome and protein data of these fungi were downloaded from Ensembl Fungi (*EnsemblFungi, 2015*).

### Identifying putative proteins contributing to pathogenicity

To identify putative genes contributing to pathogenicity in *G. morbida*, a BLASTp search was conducted for single best hits at an e-value threshold of 1e-6 or less against the PHI-base database (v3.8) (https://goo.gl/CEEVY0) that contains experimentally confirmed genes from fungal, oomycete and bacterial pathogens (*PHI-base, 2015*).

The search was performed using the same parameters for *F. solani* and *G. clavigera*. To identify the proteins that contain signal peptides, we used SignalP (v4.1) (https://goo.gl/JOe5Dh), and compared results from *G. morbida* with those from *F. solani* and *G. claviger*a (*Peterson et al., 2011*). Lastly, to find putative protein domains involved in pathogenicity in *G. morbida*, we performed a HMMER (version 3.1b2) (*Finn, Clements & Eddy, 2011*) search against the Pfam database (v28.0) (*Finn et al., 2014*) using the protein sequences as query. We conducted the same search for sequences of 17 known effector proteins, then extracted and analyzed domains common between the effector sequences and *G. morbida* (https://goo.gl/Y9IPZs).

## RESULTS AND DISCUSSION

### Data processing

A total of 28,027,726 paired-end (PE) and 41,348,578 mate-pair (MP) reads were generated with approximately 109x and 160x coverage, respectively (Table 1). Of the MP reads, 67.7% contained adapters that were trimmed using NextClip (v1.3.1). We corrected errors within the PE reads using BLESS (v0.16) at a kmer length of 21. After correction, low-quality reads (phred score < 2) were trimmed with Trimmomatic (v0.32) resulting in 99.75% reads passing. In total, 16,336,158 MP and 27,957,268 PE reads were used to construct the *de novo* genome assembly.

### Assembly features

The *G. morbida de novo* assembly was constructed with AllPaths-LG (v49414). The assembled genome consisted of 73 contigs totaling 26,549,069 bp, which is comparable to certain other Ascomycetes such as *Acremonium chrysogenum* and *Ustilaginoidea virens* with genome sizes of 28.6 and 30.2 Mbp, respectively. The largest contig length was 2,597,956 bp, and the NG50 was 1,305,468 bp. The completeness of the genome assembly was assessed using BUSCO, a tool that scans the genome for the presence of single-copy orthologous groups present in more than 90% of fungal species. Of 1,438 single-copy orthologs specific to fungi, 98% were complete in our assembly, and 4.3% were duplicated BUSCOs. Only one ortholog was missing from the genome (Table 2). We used BWA to map the unprocessed, raw MP and PE reads back to the genome to further evaluate the assembly, and 87% of the MP and 90% of the PE reads mapped to our reference genome.

### Gene annotation

The automated genome annotation software Maker v2.31.8 was used to identify structural elements in the *G. morbida* assembly generated by AllPaths-LG. Of the total 6,273 proteins that were predicted, 5,996 had protein-homology evidence in the Swiss-Prot database and only 277 (4.41%) of the total genes encoded for proteins of unknown function. Even though the total of 6,273 proteins is lower than the average number of 11,129 genes in Ascomycota, this number is within the range of the 4,657 and 27,529 coding genes within the phylum (*Mohanta & Bae, 2015*). The completeness of the functional annotations was evaluated using BUSCO, and 95% of the single copy orthologs were present in this protein set and only 7% were duplicated BUSCOs.

**Table 1 Statistics for *Geosmithia morbida* sequence data.** The values in bold are final number of reads used for assembly after quality check.

|  | Paired-end | | Mate-pair | |
| --- | --- | --- | --- | --- |
| Number of reads | 28,027,726 | **27,957,268** | 41,348,578 | **16,336,158** |
| Average insert size (bp) | 487 | | 1921 | |
| Average coverage | 109x | | 160x | |

**Table 2 *Geosmithia morbida* reference genome assembly statistics generated using QUAST (v2.3).**

| | |
| --- | --- |
| Number of sequences | 73 |
| Largest scaffold length | 2,597,956 |
| N50 | 1,305,468 |
| L50 | 7 |
| Total assembly length | 26,549,069 |
| GC% | 54.31 |
| BUSCOs completeness | 98% |

## Repetitive elements

Repetitive elements represented 0.81% of the total bases in *G. morbida*. The genome contained 152 retroelements (class I) that were mostly composed of long terminal repeats (n = 146) and 60 DNA transposons (class II). In comparison, the genomes of *G. clavigera* and *F. solani* contained 1.14 and 1.47%, respectively. *G. clavigera* possesses 541 retroelements (0.79%) and 66 DNA transposons (0.04%), whereas the genome of *F. solani* is comprised of 499 (0.54%) and 515 (0.81%) retroelements and transposons, respectively. The larger number of repeat elements in *F. solani* may explain its relatively large genome size—51.3 Mbp versus *G. clavigera's* 29.8 Mbp and *G. morbida's* 26.5 Mbp (Table 3).

## Identifying putative pathogenicity genes

We blasted the entire predicted protein set against the PHI-base database (v3.8) to identify a list of putative genes that may contribute to pathogenicity within *G. morbida*, *F. solani*, and *G. clavigera*. We determined that 1,974 genes in *G. morbida* (31.47% of the total 6,273 genes) were homologous to protein sequences in the database (Table S1). For *F. solani* and *G. clavigera*, there were 4,855 and 2,387 genes with homologous PHI-base proteins (Tables S2 and S3).

## Identifying putative secreted proteins

A search for the presence of putative secreted peptides within the protein sequences of *G. morbida*, *F. solani* and *G. clavigera* showed that approximately 5.6% (349) of the *G. morbida* sequences contained signal peptides (Table S4). Of the 349 sequences containing putative signal peptides, only 27 encoded proteins of unknown function. Roughly 8.8 and 6.9% of the proteins of *F. solani* and *G. clavigera* possess signal peptides (Tables S5 and S6). Secreted proteins are essential for host-fungal interactions and are indicative of adaptation within fungal pathogens that require an array of mechanisms

**Table 3 Repetitive elements profile for *Geosmithia morbida*, *Grosmannia clavigera*, and *Fusarium solani*.** RepeatMasker (v4.0.5) was used to generate these values. Genomic data for *F. solani* and *G. clavigera* were downloaded from Ensembl Fungi.

|  | G. morbida | G. clavigera | F. solani |
|---|---|---|---|
| Genome size | 26.5 Mbp | 29.8 Mbp | 51.3 Mbp |
| % Repetitive element | 0.81% | 1.14% | 1.47% |
| % Retroelements | 0.10% | 0.79% | 0.54% |
| % DNA transposons | 0.02% | 0.04% | 0.81% |

to overcome plant host defenses. Even though the precise means by which fungal proteins are trafficked into the host are unclear, secreted proteins are known to be essential for the translocation of fungal proteins into the host cells (*Petre & Kamoun, 2014*). For instance, race 1 strains of *Verticillium dahliae*, a common cause of vascular wilt disease in plants, secretes a protein called Ave1 that induces host immunity response suggesting this protein is crucial for virulence (*de Jonge et al., 2012*). Another example of a secreted protein is Ecp6 in fungal pathogen *Cladosporium fulvum* that prevents chitin-activated detection by the host plant (*de Jonge et al., 2010*).

### Identifying protein domains

We conducted a HMMER search against the pfam database (v28.0) using amino acid sequences for *G. morbida* and 17 effector proteins from various fungal species. For *G. morbida*, there were 6,023 unique protein domains out of a total of 43,823 Pfam hits. A total of 17 domains, which comprised 1,000 hits, were shared between *G. morbida* and known effector proteins. The three most common protein domains in *G. morbida* with a putative effector function belonged to short-chain dehydrogenases (n = 111), polyketide synthases (n = 94) and NADH dehydrogenases (n = 86). The HMMER *G. morbida* and effector proteins output files can be found in Tables S7 and S8.

### CONCLUSION

This work introduces the first genome assembly and analysis of *Geosmithia morbida*, a fungal pathogen of the black walnut tree that is vectored into the host via the walnut twig beetle. The *de novo* assembly is composed of 73 scaffolds totaling in 26.5 Mbp. There are 6,273 predicted proteins, and 4.41% of these are unknown. In comparison, 68.27% of *F. solani* and 26.70% of *G. clavigera* predicted proteins are unknown. We assessed the quality of our genome assembly and the predicted protein set using BUSCO, and found that 98 and 95% of the single copy orthologs specific to the fungal lineage were present in both, respectively. These data are indicative of our assembly's high quality and completeness. Our BLASTp search against the PHI-base database revealed that *G. morbida* possesses 1,974 genes that are homologous to proteins involved in pathogenicity. Furthermore, *G. morbida* shares several domains with known effector proteins that are key for fungal pathogens during the infection process.

*Geosmithia morbida* is one of only two known fungal pathogens within the *Geosmithia* genus (*Lynch et al., 2014*). The genome assembly introduced in this study can be

leveraged to explore the molecular mechanisms behind pathogenesis within this genus. The putative list of pathogenicity genes provided in this study can be used for future comparative genomic analyses, knock-out, and inoculation experiments. Moreover, genes unique to *G. morbida* may be utilized to develop DNA sequence-based tools for detecting and monitoring ongoing and future TCD epidemics.

### Funding
Partial funding was provided by the New Hampshire Agricultural Experiment Station. Funding was also provided by the USDA Forest Service, Forest Health and Protection. The funders had no role in study design, data collection and analysis, decision to publish, or preparation of the manuscript.

### Competing Interests
The authors declare no competing interests. Mention of a trademark, proprietary product, or vendor does not constitute a guarantee or warranty of the product by the U.S. Department of Agriculture and does not imply its approval to the exclusion of other products or vendors that also may be suitable.

### Author Contributions
- Taruna A. Schuelke conceived and designed the experiments, performed the experiments, analyzed the data, wrote the paper, prepared figures and/or tables, reviewed drafts of the paper.
- Anthony Westbrook analyzed the data, reviewed drafts of the paper.
- Kirk Broders conceived and designed the experiments, contributed reagents/materials/ analysis tools, wrote the paper, reviewed drafts of the paper, conceived funding.
- Keith Woeste conceived and designed the experiments, contributed reagents/materials/ analysis tools, wrote the paper, reviewed drafts of the paper, conceived funding.
- Matthew D. MacManes conceived and designed the experiments, analyzed the data, reviewed drafts of the paper.

### DNA Deposition
The following information was supplied regarding the deposition of DNA sequences:

The pep files for F. solani (v2.0.29) and G. clavigera (kw1407.GCA_000143105.2.30) were downloaded from FungalEnsembl.

### Data Availability
The raw reads and assembled sequences reported in this manuscript are available at European Nucleotide Archive under Project Number PRJEB13066. The in silico generated transcript and protein files are located at Dryad (doi:10.5061/dryad.d18mc). The code is available at https://github.com/macmanes-lab/Geosmithia_manuscript.
## Supplemental Information

Supplemental information for this article can be found online at http://dx.doi.org/10.7717/peerj.1952#supplemental-information.

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
