# Peer review of "De novo* genome assembly of *Geosmithia morbida*, the causal agent of thousand cankers disease"

_PeerJ, doi:10.7717/peerj.1952_

## Round 0.1 · original submission · Major Revisions

Please go through all suggested comments and revise your manuscript before we can considery accepting your manuscript.

Reviewer 1 ·

Basic reporting

The manuscript is well written and the background information relevant for the study. I would've liked to see a picture of Geosmithia morbida embedded in the manuscript. I did not see any repository accession numbers, but at some point the authors should submit their genome/annotations/raw reads to a repository. Other than that, I only have few editing suggestions that can be found in the General comments section.

Experimental design

The research questions were well defined in the introduction, and the methods appropriate and well described. Overall, the data was well analyzed.

Validity of the findings

The results seem robust and conclusive.

Additional comments

General comments:

Introduction

L48: Wording suggestion: “.. beneficial for the study of emerging..” .

L51: Wording suggestion: “Fungal pathogens are capable of rapid evolution in order..:.

L52: “... data is useful to identify...”.

L56: I think its better to used the accepted name, Zymoseptoria tritici, in line and Mycosphaerella graminicola, the synonym, in parenthesis.

L58: S1 and S2 are now known as Zymoseptoria pseudotritici and Z. ardabiliae. See http://www.mycologia.org/content/104/6/1397.full

L62: “Using phylogenomic analyses , they show that..”.

L64: “as well as the importance of genomics and bioinformatics in..”

L66: “bark beetles in the family Scolytinae”.

L67: “With climate change, beetles and their fungal symbionts can invade..”.

L72: is it “julandis” or “juglandis”?

L74: “mortality, triggered … infections, ”.

L75: instead of “and”, “while”?

L76: “agent of the ..”.

L85: “of the genera” instead of “of genus”.

L95: You are saying “complex” in the same sentence twice. “The ecological complexity in this vector-host-pathogen system..”. I would remove “within the fungal kingdom”, its redundant.

L98: “enable us to explore genomic features..”? You need much more than an assembled reference genome to identify genes involved with pathogenicity.

L103: Again, you need more data to study the evolution. But you can say, “to have a better understanding of the genomic composition/features/structure in the Geosmithia genus”.

Methods

L116: Was this sequenced on a single Illumina lane? Should be stated. Please also state the read length.

L119: The first sentence is does not provide much information.

L124: Can you state which “assembly tools and parameters”?

L132: Another option here is CEGMA, how does it compare to BUSCO?

L139/141: I think you should move reference 29 to line 139, this way the reader can quickly reference the paper associated with the software.

L140: “protein-homology evidence”.

Results

L172: Please add the PE and MP contractions to where “pair end” and “mate pair” where first mentioned in the text.

L172: These are not really “forward and reverse reads”, like in primer pairs. They are “pair ended” reads and you are already using this term. Just simply use “reads”.

L173: Comma after “coverage”.

L177: “used for de novo..”

L185: This should be “final number of reads used for assembly after QC”.

L188: This sentence is redundant.

References

L340-343, 347-348, 379-389: There seems to be a line break problem here.

·

Basic reporting

This brief article is well-written and provides appropriate rationale on the importance of sequencing the fungal tree pathogen. The authors also provide a sufficient literature overview of the importance of the work. It appears that the authors have followed guidelines for submission and have provided links for supplemental information concerning the genome.

Experimental design

Protocols for genome data processing are explained and appear appropriate except for a few minor clarifications I suggest (see general comments).

Validity of the findings

The findings are valid and should provide a benchmark for further genomic studies on Geosmithia morbida.

Additional comments

I suggest that the authors make some minor modifications in the text as outline below.

Page 3 line 88. Possibly mention the occurrence of TCD in Europe?
Page 4 line 107. I think this isolate should be deposited in ATCC or CBS because it is the first to be sequenced.
Page 7 line 179. How does the genome size of G. morbida compare to other ascomycetes?
Page 8 line 201. I am a bit confused here. Are the 6,273 predicted proteins based on the Swiss-Prot database only or does it refer to all predicted genes, i.e. were there predicted genes with no match to proteins in the Swiss-Prot database and if so, what is that total number? If the 6,273 is for all predicted genes, then that number seems low compared to other ascomycetes that have been sequenced.
Page 9 line 227. I suggest you add the term putative here, even though you have it in the heading. There are many examples of organisms with proteins having signal peptides that have not been shown to be secreted. Possibly provide a few references here as well.
Page 9 line 234. Maybe elaborate briefly with a few references?

---

## Round 0.2 · accepted · Accept

We have good news for you. The reviewer's points are answered and so this article is deemed suitable for publication.

Reviewer 1 ·

Basic reporting

No Comments

Experimental design

No Comments

Validity of the findings

No Comments